

# Measurements of white-to-white corneal diameter and anterior chamber parameters using the Pentacam AXL wave and their correlations in the adult Saudi population

Wafa M. Alotaibi[1], Naveen Challa[2], Saif Hassan Alrasheed[2] and Rahaf Nasser Abanmi[1]

[1] Department of Optometry and Visual Sciences, College of Applied Medicine Sciences, King Saud University, Riyadh, Saudi Arabia
[2] Department of Optometry, College of Applied Medical Sciences, Qassim University, Buraydah, Saudi Arabia

## ABSTRACT

**Background:** Assessment of eye parameters such as the white-to-white (WTW) corneal diameter, anterior chamber depth (ACD), and anterior chamber angle (ACA) are essential for preoperative evaluation of refractive surgery and diagnosing and managing some ocular diseases.

**Objective:** To report the distribution and association between white-to-white corneal diameter and anterior chamber parameters in the Saudi adult population.

**Methods:** Cross-sectional prospective study consisting of 82 Saudi adults. White-to-white corneal diameter, anterior chamber angle, anterior chamber depth, and axial length were collected from healthy subjects using the Pentacam AXL Wave in a tertiary eye care setting.

**Results:** Mean white-to-white corneal diameter, anterior chamber angle, anterior chamber depth, and axial length were $11.95 \pm 0.39$ mm, $37.94 \pm 5.01°$, $2.97 \pm 0.31$ mm and $23.93 \pm 0.97$ mm, respectively. A significant moderate correlation was found between white-to-white corneal diameter and anterior chamber angle ($r = 0.31$, $p \leq 0.01$), anterior chamber depth ($r = 0.49$, $p \leq 0.01$), and axial length ($r = 0.50$, $p \leq 0.01$) and between anterior chamber angle and anterior chamber depth ($r = 0.71$, $p \leq 0.01$). Furthermore, age was moderately correlated with anterior chamber angle ($r = 0.44$, $p$ 0.01) and anterior chamber depth ($r = 0.39$, $p$ 0.01) and weakly correlated with white-to-white corneal diameter ($r = 0.17$, $p = 0.12$). Males had a significantly ($p < 0.01$) higher white-to-white corneal diameter ($12.12 \pm 0.38$ mm) than females ($11.84 \pm 0.36$.mm).

**Conclusion:** Reference values for white-to-white corneal diameter and anterior chamber parameters would help eye care professionals choose the right lenses for cataract and refractive surgeries as well as in diagnosing glaucoma and corneal disorders.

Corresponding author
Saif Hassan Alrasheed,
s.rasheed@qu.edu.sa

## INTRODUCTION

Assessment of eye parameters such as the white-to-white (WTW) corneal diameter, anterior chamber depth (ACD), and anterior chamber angle (ACA) are essential for preoperative evaluation of refractive surgery and diagnosing and managing some ocular diseases (*Fernández-Vigo et al., 2016*; *Reinstein, Gobbe & Archer, 2012*; *Sedaghat et al., 2017*). The white-to-white corneal diameter is the horizontal measurement between the temporal and nasal limbus, which corresponds to the outer edge of the cornea. The WTW distance is important for determining the appropriate size of intraocular lenses (IOLs) during cataract surgery. It also aids in the assessment of eye size for certain refractive procedures like phakic intraocular lens implantation or intracorneal ring segment (ICR) placement (*Wei et al., 2021*). The anterior chamber depth (ACD) refers to the distance between the cornea and the lens in the front part of the eye. The ACD measurement is crucial for evaluating the risk of angle-closure glaucoma, assessing the suitability for certain intraocular procedures, and determining the appropriate IOL power for cataract surgery (*Ning et al., 2019*). The anterior chamber angle (ACA) is the angle formed between the cornea and the iris at the periphery of the eye. The ACA assessment helps identify abnormalities and conditions such as angle-closure glaucoma and also helps in determining the appropriate management strategies, such as laser peripheral iridotomy or surgical intervention (*Riva et al., 2020*). Portions of this text were previously published as part of a preprint https://doi.org/10.21203/rs.3.rs-4016989/v1.

The incorrect size of the IOL or ICR may increase risk of complications, such as endothelial cell damage, corneal edema, reduced vision, and potential long-term complications (*Vaiciuliene et al., 2022*; *Hofling-Lima et al., 2004*). An abnormal WTW corneal diameter can indicate various corneal diseases such as micro-cornea and microphthalmos (*Xu et al., 2021*). Therefore, it is important to recognize these conditions before ocular surgery. Recent studies have shown that the WTW corneal diameter and ACD are important parameters in some formulas used to calculate the power of the IOL for cataract surgery (*Stopyra, Langenbucher & Grzybowski, 2023*).

Previous studies investigated the anterior chamber parameters in different populations (*Alrajhi, Bokhary & Al-Saleh, 2018*; *Hashemi et al., 2010*; *Rüfer, Schröder & Erb, 2005*). However, the results of these studies may not apply to the Saudi population due to ethnic differences. From the review of literature, little information is known regarding the correlation between the WTW corneal diameter, ACD, and ACA within a Saudi Arabian cohort. Such parameters are also critical for cataract and refractive surgeries, and glaucoma diagnostics especially since the prevalence of myopia and diabetes is known to be high in Saudi Arabia and thus these factors might affect these metrics differently in Saudi Arabia compared to Western and Asian populations. Therefore, this study aims to determine the distribution of the WTW corneal diameter, ACD, and ACA among the adult Saudi population and the correlations among these parameters.

## MATERIALS AND METHODS

### Study design

This prospective cross-sectional study was conducted at King Saud University, Riyadh, Saudi Arabia from January 2022 to April 2022 and involved healthy young male and female individuals visiting optometry clinics.

### Sample size

The study used a non-probability sampling method to select participants from the optometry clinic at King Saud University. It included a total of 82 healthy individuals, both male and female, ranging in age from 20 to 60 years.

### Inclusion criteria

Participants with prior eye surgery and ocular conditions such as diabetic retinopathy, corneal opacity, cataracts, and nystagmus that could impact measurement accuracy were not included. Pregnant women and smokers were also excluded.

### Ethical approval

The study followed ethical guidelines from King Saud University Medical City's ethical committee (approval number E-21-6448) and adhered to the principles of the Declaration of Helsinki. Written informed consent was obtained from all participants prior to performing the eye examination and after they were given a clear explanation of the study's purpose. Confidentiality was maintained to protect the anonymity of all individuals involved in the data collection.

### Data collection procedures

Before performing WTW corneal diameter measurements and anterior chamber parameter readings, participants provided comprehensive demographic information, including age, sex, refractive errors, medical history, and reported symptoms. Subsequently, all individuals underwent ocular examinations using specific equipment, such as a slit lamp (Haag-Streit BQ 900) to assess any ocular abnormalities and an autorefractometer (Topcon KR-1 Autorefractor/Keratometer) to determine refractive error. Eligible subjects then underwent measurements of ACA, ACD, and WTW corneal diameter using the OCULUS Pentacam AXL Wave (Oculus, Inc., Arlington, WA, USA).

The Pentacam AXL Wave is the relatively latest instrument for reliably measuring and analysing the anterior segment of the eye (*Balparda et al., 2022*). It combines anterior segment tomography with integrated axial length measurement. The Pentacam AXL Wave has a quality specification feature (QS) that assesses the scan quality and indicates an acceptable image with an 'OK' result (*OCULUS*). The Pentacam AXL wave was chosen in our study because of the very high precision with which the anterior segment is captured and the measurement of WTW corneal diameter, ACD, and ACA with minimal operator dependency, and therefore superior to manual or older automated methods.

To ensure reproducibility, the device underwent regular calibration as per manufacturer guidelines. The imaging was conducted in a dark room, with participants positioned in

front of the camera and resting the chin and forehead on the frame. The image was adjusted using a joystick until the corneal surface appeared on the monitor, centring the pupil and corneal apex using the marker. Participants were instructed to blink to maximize tears and then to focus on the fixation target for 2 s during the scanning process. All measurements, including ACA, ACD, and WTW corneal diameter, were performed automatically, and the mean of three readings was recorded for each participant who met the inclusion criteria. The examinations were performed on the right eye between 8:00 and 11:00 AM to minimize the effects of diurnal variations in AL (*Alanazi et al., 2021*; *Ulaganathan et al., 2019*). The average of three measurements for the WTW corneal diameter, ACD, and ACA was computed for each subject and used for final analysis. The entire examination was conducted on a single eye; specifically, the right eye by a trained examiner. Examiner underwent extensive training, including certification from the manufacturer and multiple practice sessions before data collection commenced. These measures ensured consistency and minimized inter-operator variability.

## Data analysis

The collected data were entered into an Excel spreadsheet and analysed using IBM SPSS Statistics for Windows, version 25 (IBM Corp., Armonk, N.Y., USA). The Shapiro-Wilk test was employed to assess for normality of data. If the data exhibited a normal distribution, descriptive statistics were used to summarize the ocular biometric measurements. Pearson's correlation coefficient was utilized to assess the relationships among the ACD, ACA, and WTW corneal diameter. Correlation coefficient (r) value under 0.1 is reported as very weak correlation, 0.1 to 0.30 is reported as weak correlation, 0.3 to 0.6 is reported as moderate correlation and above 0.6 is reported as high correlation. A value of $p < 0.05$ was considered statistically significant.

## RESULTS

The sample consisted of 82 Saudi adults, including 51 females and 31 males. The age range of the participants was 20–60 years with a mean of 28.7 ± 11.19 years. The mean spherical equivalent refractive error in the sample ranged from −4.125 D to +3.625 D, as shown in Table 1.

### The distribution of the WTW corneal diameter and the AC parameters

The WTW corneal diameter (skewness = −0.11, kurtosis = −0.39, KS $p = 0.67$), ACD (skewness = 0.21, kurtosis = 0.30, KS $p = 0.88$), and ACA (skewness = 0.11, kurtosis = 0.39, KS $p = 0.07$) exhibited a normal distribution. Figure 1 shows the distribution curves of the WTW corneal diameter, ACD, and ACA of the study sample. On the other hand, spherical equivalent (KS $p < 0.01$) and age (KS $p < 0.01$) were not normally distributed.

The descriptive statistics of the WTW corneal diameter, ACA, ACD, and AL are presented in Table 1. The mean WTW corneal diameter, ACA, ACD, and AL were 11.95 ± 0.39 mm, 37.94 ± 5.01°, 2.97 ± 0.31 mm and 23.93 ± 0.97 mm, respectively. Correlations between the WTW corneal diameter, anterior chamber parameters, and AL are presented in Table 2.

**Table 1 A comparative analysis of ocular biometric parameters between males and females.** Mean values, standard deviations, 95% confidence intervals, and ranges are provided for each parameter. Males exhibited significantly larger white-to-white corneal diameter (WTW) and axial length (AL) compared to females. Descriptives of WTW corneal diameter and Anterior chamber parameters of male and females in study sample. *p*-value of ANOVA among the both genders is in the last column of the table.

| | | N | Mean | Std. Dev | 95% Confidence interval for mean | | Min | Max | *p*-value |
|---|---|---|---|---|---|---|---|---|---|
| | | | | | Lower bound | Upper bound | | | |
| Mean ACA | Male | 31 | 39.06 | 5.54 | 37.03 | 41.09 | 28.90 | 50.40 | 0.12 |
| | Female | 51 | 37.26 | 4.58 | 35.97 | 38.55 | 28.20 | 44.93 | |
| | Total | 82 | 37.94 | 5.01 | 36.84 | 39.04 | 28.20 | 50.40 | |
| Mean ACD | Male | 31 | 3.05 | 0.34 | 2.93 | 3.17 | 2.23 | 3.64 | 0.06 |
| | Female | 51 | 2.91 | 0.29 | 2.83 | 3.00 | 2.16 | 3.49 | |
| | Total | 82 | 2.97 | 0.31 | 2.90 | 3.03 | 2.16 | 3.64 | |
| Mean WTW | Male | 31 | 12.12 | 0.38 | 11.98 | 12.26 | 11.17 | 12.83 | 0.01* |
| | Female | 51 | 11.84 | 0.36 | 11.74 | 11.94 | 11.03 | 12.63 | |
| | Total | 82 | 11.95 | 0.39 | 11.86 | 12.03 | 11.03 | 12.83 | |
| Mean AL | Male | 31 | 24.32 | 0.77 | 24.03 | 24.60 | 22.75 | 25.63 | 0.01* |
| | Female | 51 | 23.70 | 1.01 | 23.41 | 23.98 | 21.69 | 26.73 | |
| | Total | 82 | 23.93 | 0.97 | 23.72 | 24.14 | 21.69 | 26.73 | |
| Mean S_EQUV | Male | 31 | −0.77 | 1.32 | −1.26 | −0.28 | −3.75 | 2.75 | 0.35 |
| | Female | 51 | −1.07 | 1.41 | −1.47 | −0.67 | −4.125 | 3.625 | |
| | Total | 82 | −0.96 | 1.38 | −1.26 | −0.65 | −4.00 | 3.625 | |

Results show that there was a significant moderate correlation between the WTW corneal diameter and ACA ($r = 0.31$, $p \leq 0.01$) (Fig. 2), WTW corneal diameter and ACD ($r = 0.49$, $p \leq 0.01$) (Fig. 2), and WTW corneal diameter and AL ($r = 0.50$, $p \leq 0.01$) (Fig. 3). A significantly high correlation ($r = 0.71$, $p \leq 0.01$) (Fig. 3) was observed between the ACA and ACD.

**Age:** A significant positive moderate correlation was observed between age and ACA ($r = 0.44$, $p \leq 0.01$) and ACD ($r = 0.39$, $p \leq 0.01$); however, a weak non-significant correlation was found between age and the WTW corneal diameter ($r = 0.17$, $p = 0.12$).

**Sex:** The mean WTW corneal diameter in males ($12.12 \pm 0.38$ mm) was significantly larger than females ($11.84 \pm 0.36$ mm) ($p \leq 0.01$) as shown in Table 1. The mean ACA and ACD were slightly higher in males than females but not significantly different (ACA, $p = 0.12$, ACD $p = 0.06$) (Table 1).

### Refractive error

The spherical equivalent had a very weak negative correlation with the WTW corneal diameter ($r = -0.09$, $p = 0.60$) but showed a very weak correlation with ACA ($r = 0.030$, $p = 0.79$) and ACD ($r = 0.02$, $p = 0.88$).

### Axial length

Current study shows a significant positive moderate correlation between AL and WTW corneal diameter ($r = 0.50$, $p < 0.01$), AL and ACA ($r = 0.32$, $p \leq 0.01$) and, AL and ACD

a)

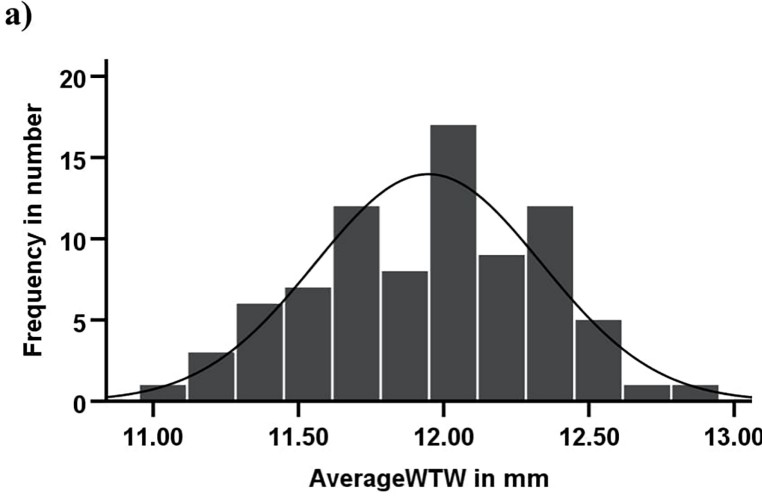

b)

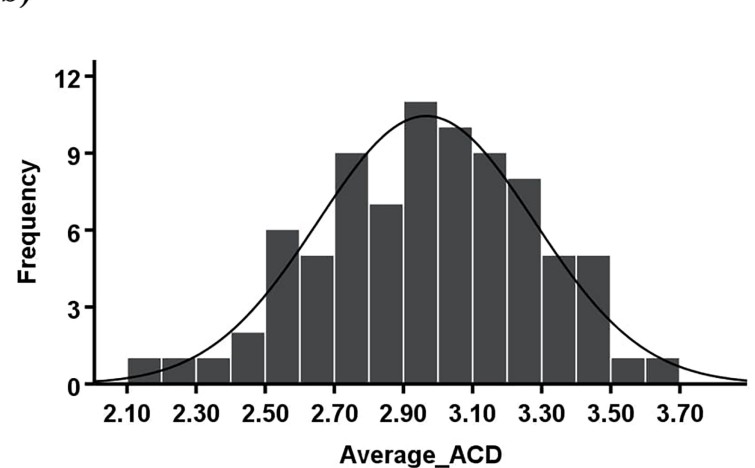

c)

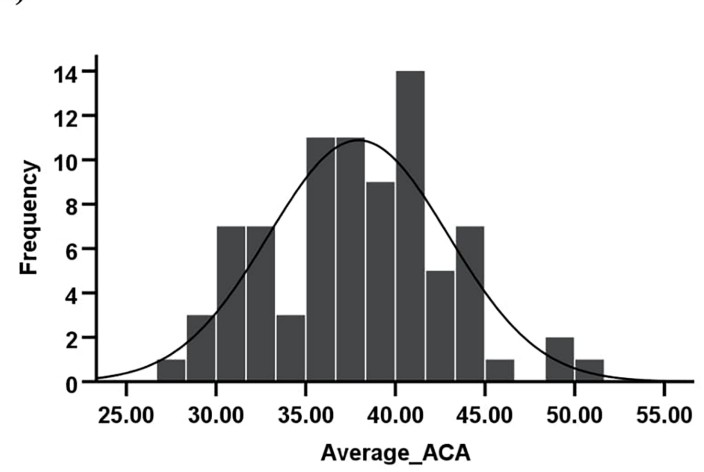

**Figure 1 Histograms showing the distribution of key ocular biometric parameters in the study population.** (A) WTW (white to white) corneal diameter follows an approximately normal distribution, peaking around 12.0 mm. (B) Anterior chamber depth (ACD) shows a slightly skewed normal distribution,

**Figure 1** (continued)
with most values between 2.5 and 3.5 mm, peaking around 2.9–3.1 mm. (C) Anterior chamber angle (ACA) has a broader distribution (25–55°), with most values between 35° and 45°, peaking around 42°. Solid lines represent the fitted normal distribution curves.

**Table 2 Shows the correlations between ocular parameters and age, as well as inter-relationships among parameters.** Age signficantly negatively correlated with anterior chamber depth (ACD) and anterior chamber angle (ACA). White-to-white (WTW) corneal diameter was significantly positively correlated with ACD, ACA, and axial length (AL). ACD also signficantly positively correlated with ACA and AL, while ACA showed a positive correlation with AL. Correlations between age, WTW corneal diameter, ACD, ACA, and AL.

| Parameter | Average WTW | Average ACD | Average ACA | Average AL | Average SE |
|---|---|---|---|---|---|
| Age | −0.17 | −0.39** | −0.44** | −0.03 | −0.15 |
| *P* value | 0.12 | 0.01 | 0.01 | 0.78 | 0.17 |
| Average WTW | | 0.49** | 0.31** | 0.50** | 0.09 |
| *P* value | | 0.001 | 0.001 | 0.001 | 0.60 |
| Average ACD | | | 0.71** | 0.46** | 0.02 |
| *P* value | | | 0.001 | 0.01 | 0.88 |
| Average ACA | | | | 0.32** | 0.03 |
| *P* value | | | | 0.001 | 0.79 |

**Note:**
** indicates significance level < 0.01.

(r = 0.46, $p \leq 0.01$) suggesting longer the axial length, larger the WTW corneal diameter, deep ACA, and longer ACD.

Multiple linear regression was used to determine the effect of confounding factors such as age, gender and refractive error on WTW corneal diameter, ACA, ACD. The results (Table 3) show that gender, age and refractive error together account for 16.1% of variance in WTW corneal diameter, 23.5% of variance in ACD, and has 28.5% of variance in ACA. Moreover, results shows that female gender was significantly negatively influenced WTW corneal diameter (beta = −0.34, $p < 0.05$). Age significantly negatively influenced ACD (beta = −0.35, $p < 0.01$) and ACA (beta = −0.41, $p < 0.01$). There was a negative correlation between spherical equivalent refractive error and ACD (beta = −0.25, <0.01) and ACA (beta = −0.28, <0.05).

## DISCUSSION

The distribution of the WTW corneal diameter and its relationship to ACD characteristics in the adult Saudi population was investigated in this study. The mean WTW corneal diameter in our study was 11.95 mm, with 95% of the eyes ranging from 11.17 to 12.70 mm. The mean ACD in the current study sample was 2.97 mm, with 95% of the eyes

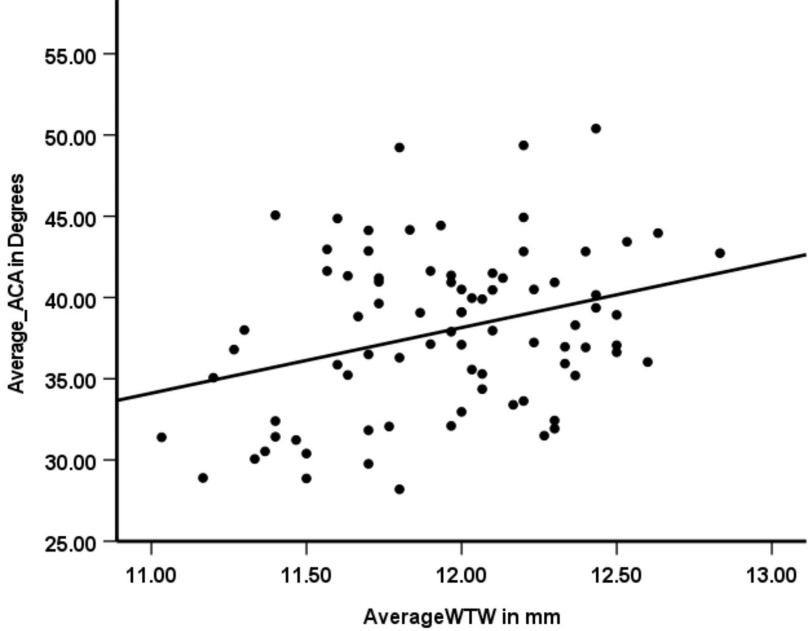

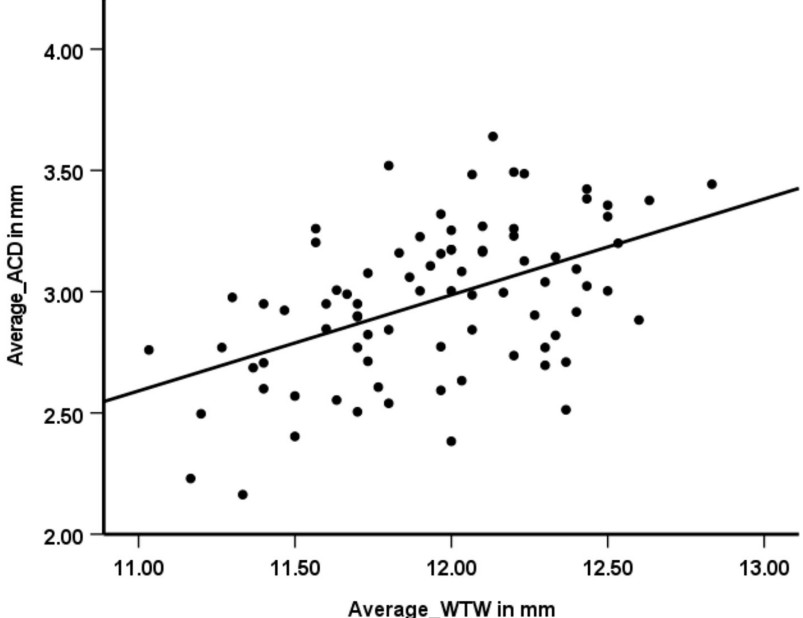

**Figure 2 Scatter plots showing the relationships between white-to-white (WTW) corneal diameter and anterior chamber parameters.** (A) ACA and WTW corneal diameter demonstrate a weak positive correlation, indicating that larger corneal diameters may be associated with slightly wider anterior chamber angles. (B) ACD and WTW corneal diameter show a moderate positive correlation, suggesting that a larger WTW is linked to a deeper anterior chamber. The solid lines indicate the linear regression trend.

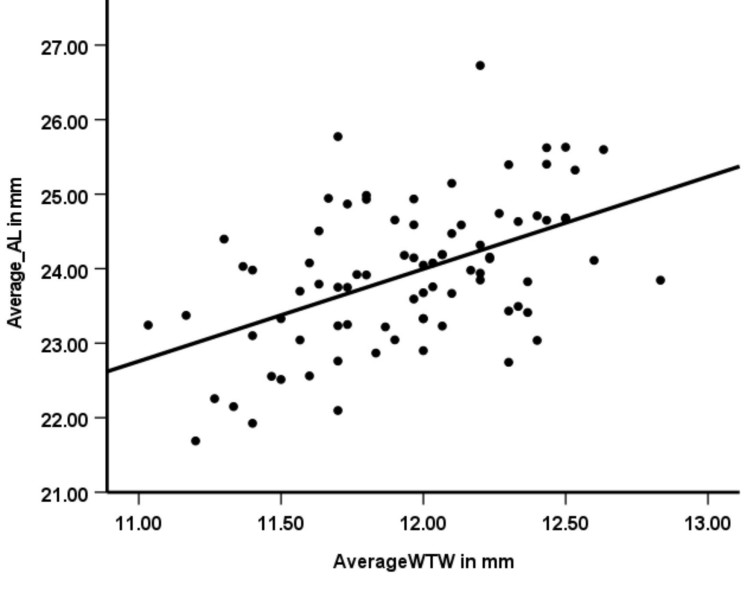

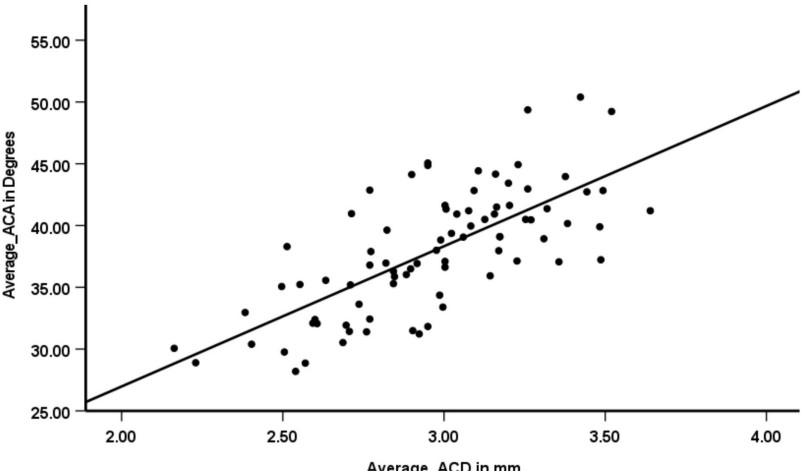

**Figure 3 Scatter plots showing correlations between anterior chamber parameters and other ocular biometric measurements.** (A) Axial length (AL) and white-to-white (WTW) corneal diameter show a positive correlation, indicating that larger corneal diameters are associated with longer axial lengths. (B) Anterior chamber angle (ACA) and anterior chamber depth (ACD) exhibit a strong positive correlation, suggesting that deeper anterior chambers tend to have wider angles. The solid lines indicate the linear regression trend.

having an ACD between 2.36 and 3.57 mm. Similarly, the average ACA was 37.94°, and 95% of the eyes had an ACA between 28.14° and 47.73°.

The typical range of the WTW corneal diameter has been the subject of numerous studies, with variable values depending on the study population and WTW measurement tool. The mean WTW corneal diameter in the current study is higher than that reported in Chinese (*Xu et al., 2021*), Iranian (*Hashemi et al., 2010*), Indian (*Singh et al., 2019*), German (*Baumeister et al., 2004*), and Australian (*Fotedar et al., 2010*) groups but comparable to that of Egyptian adults (*Singh et al., 2019*). These differences in WTW corneal diameter among the different regions may be attributed to genetic and

**Table 3 Multiple regression showing the model fit for white-to-white (WTW) corneal diameter, anterior chamber depth (ACD), and anterior chamber angle (ACA) after adjusting for the confounders.** β values suggest the percentage of variation on measured parameters caused by a confounder. Gender had a significant effect on WTW corneal diameter and ACA. Spherical equivalent significantly affected ACD but not WTW or ACA.

| Dependent variable | Independent variable | $R^2$ (Model) | β value | P-value |
|---|---|---|---|---|
| White to white corneal diameter | Age | 0.161 | −0.155 | 0.161 |
| | Gender | | −0.344 | 0.002 |
| | Spherical equivalent | | −0.016 | 0.879 |
| Anterior chamber depth | Age | 0.235 | −0.354 | 0.001 |
| | Gender | | −0.107 | 0.306 |
| | Spherical equivalent | | −0.249 | 0.018 |
| Anterior chamber angle | Age | 0.285 | −0.411 | 0.000 |
| | Gender | | −0.048 | 0.637 |
| | Spherical equivalent | | −0.281 | 0.006 |

**Table 4 Difference in WTW corneal diameter among the different races.**

| Study | Race/Ethnicity | Number of eyes | Method of measurement | Age group | WTW corneal diameter (mm) |
|---|---|---|---|---|---|
| Xu et al. (2021) | Chinese | 4,416 | Pentacam | >18 years | 11.65 ± 0.38 |
| Hashemi et al. (2010) | Iranians | 800 | Orbscan | >14 years | 11.68 |
| Singh et al. (2019) | Indians | 650 | IOL master | 10–80 years | 11.79 ± 0.05 |
| Alanazi et al. (2021) | Germans | 100 | IOL master | 20–79 years | 12.02 ± 0.38 |
| | | | Orbscan | | 11.78 ± 0.43 |
| | | | Surgical caliper | | 11.91 ± 0.71 |
| Ulaganathan et al. (2019) | Australians | 1,952 | IOL master | >59 years | 12.06 |
| Current study | Arabs | 82 | Pentacam AXL | >18 years | 11.95 ± 0.39 |

environmental determinants of anatomy in that specific region. These results indicate that the intraocular lens power calculation should be tailored to population specific biometric baselines Table 4 provides a summary of the mean WTW corneal diameter from studies utilizing different methods of measurements.

A study comparing the WTW distance using the Pentacam and Zeiss IOLMaster (Carl Zeiss Meditec AG, Germany) systems in Egyptian adults has shown that the Pentacam tend to estimate a significantly larger WTW distance compared to that of the IOLMaster ($p < 0.001$) due to different modalities used in these devices (Elkateb & Swelem, 2016). Given that Pentacam was utilized in this investigation to assess the WTW corneal diameter, we think the results of the WTW corneal diameter in our study may be larger than those of the other studies that used the instruments other than Pentcam.

In addition to race, the WTW corneal diameter values can be influenced by sex, age, refractive error, and AL. Studies on sex found a considerable difference in the WTW distance between males and females (Hashemi et al., 2010; Rüfer, Schröder & Erb, 2005;

*Balparda et al., 2022*; *OCULUS*; *Alanazi et al., 2021*; *Ulaganathan et al., 2019*; *Singh et al., 2019*; *Baumeister et al., 2004*; *Fotedar et al., 2010*; *Elkateb & Swelem, 2016*; *Smith, 1890*), but another study (*Rüfer, Schröder & Erb, 2005*) found no difference. A significant ($p < 0.01$) difference in the WTW corneal diameter was observed between males and females in the current study. This difference may be attributed to the fact that male eyeballs are generally slightly larger than those of females. Additionally, males tend to have larger orbital cavities, which could contribute to the observed variation in WTW corneal diameter between the two sexes (*Patra et al., 2021*).

Research on the relationship between WTW corneal diameter and age has yielded mixed results. While some studies (*Hashemi et al., 2010*; *Rüfer, Schröder & Erb, 2005*; *Gharaee et al., 2014*) have found a significant negative correlation, another study (*Chen & Osher, 2015*) found no correlation between age and WTW corneal diameter, which is consistent with the results of the current study. Studies have demonstrated a statistically significant positive correlation between the WTW corneal diameter and AL (*Hashemi et al., 2015*; *Park et al., 2010*). Consistent with these studies, the current study also found a significant moderate correlation between AL and WTW corneal diameter. This is an expected finding from our study. The eye develops as a cohesive system, where an increase in axial length is typically accompanied by proportional growth in other dimensions, such as the WTW corneal diameter. This coordinated development helps in emmetropisation of the eye.

The findings of the current study indicate that there is a relatively positive correlation between the ACD and WTW corneal diameter ($r = 0.49$, $p \leq 0.01$) and the ACA and WTW corneal diameter ($r = 0.31$, $p < 0.01$). Previous studies from the Chinese (*Xu et al., 2021*) and Indian (*Singh et al., 2019*) populations found a similar association.

The mean ACD measurement for the current study sample was 2.97 mm. *Osuobeni (1999)* conducted a study among Saudi adults using ultrasound and reported a mean ACD of 3.2 mm, which is higher than that of the current study. The possible difference in ACD between both studies may be due to the different instruments used. A study done by *Mashige & Oduntan (2017)* measured the ACD values using Nidek US-500 Echoscan in black Africans and compared these values with existing literature in Asians, Europeans and other races and found that ACD in Africans is shallower than the European counterpart. They reported that Europeans had the highest ACD (mean ACD 3.20–3.48 mm), followed by Americans (ACD range 3.21–3.45 mm), Africans (ACD of 3.21 mm), and Asians (ACD range 2.62–3.2 mm). This variance may be due to the use of different instruments and racial disparities. Different instruments used in various studies operate on different principles, similarly different races have different anatomical features that lead to the difference in ACD measurements. People with shallow ACD may be more susceptible to angle-closure glaucoma, which may be the reason this condition is more common in Asians than in Caucasians.

The use of the implantable collamer lens (ICL) is considered one of the safest, most reliable, and reversible phakic surgical procedures. Although the ICL procedure is safe, there are several possible side effects associated with the surgery including inflammation in the anterior chamber, higher intraocular pressure, lens opacification, and ICL rotation.

Most of the complications arise from incorrect ICL vaulting results from choosing the wrong ICL size. Preoperative ACD and WTW assessments are important measurements that would help in choosing the correct ICL size. An appropriately sized ICL can significantly aid in achieving an optimal vault after surgery. Determining the normal ranges of ACD and WTW is crucial in that they serve as references for ICL sizing. According to a recent study by *Tang et al. (2023)*, it is best to use the manufacturer's nomogram in conjunction with the Pentacam system to measure the WTW distance when selecting an ICL size. The Pentacam was employed in the current investigation to measure the WTW distance, which would help to better understand the typical distribution of the WTW corneal diameter for future reference values in the Saudi population undergoing ICL surgery.

Currently, Vision ICL (Staar Surgicals, Monrovia, CA, USA) is not available for people with an ACD less than 2.8 mm (*Xu et al., 2021*). In the current study, 25 subjects (30.5%) had an ACD of less than 2.8 mm, out of which 13 were myopes and 12 were hyperopes. All these subjects were not suitable for Vision ICL surgery.

Our findings align with previous research indicating that male subjects generally have larger WTW corneal diameters and longer axial lengths than females (*Hashemi et al., 2012*). These anatomical differences are clinically significant, as they impact the selection of intraocular lenses and the sizing of phakic intraocular lenses for refractive procedures.

Additionally, we observed a significant negative correlation between age and anterior chamber parameters, consistent with earlier studies suggesting that aging contributes to a gradual narrowing of the anterior chamber, which may increase the risk of angle-closure glaucoma (*Sun et al., 2012*). The relationship between spherical equivalent and ACD and ACA is particularly relevant for refractive surgery planning, as these factors are crucial in determining the suitability of lens implantation and also for safe postoperative outcomes (*Tang et al., 2023*).

Our findings highlight the importance of establishing population-specific biometric reference values, particularly in regions such as Saudi Arabia, where the high prevalence of myopia may influence ocular dimensions differently than in Western or Asian populations.

Although the current study has provided some valuable insights in understanding the relationship between WTW corneal diameter and anterior chamber parameters, it has a limitation of sample size and the age of the participants skewed towards younger ages, may affect the correlation between age and other parameters. Apart from this inter operator variability and environmental factors such as lighting could have introduced small bias in the measurements. However, lighting effects were reduced by conducting pentacam measurements in particular time of the day in a dark room. Although the sample size was limited, these findings provide some baseline biometric data for the Saudi population. Future research should use larger, more diverse samples and longitudinal data to track changes among groups. Replicating the same study in multiple populations would validate these results, which could then be contributed to the global ocular biometry database. One more limitation of our study is the absence of a control group with specific eye conditions like glaucoma or keratoconus, which affects how broadly our findings apply to individuals

 

with these diseases. Research has shown that people with narrower anterior chamber angles and shallower anterior chamber depths are at risk of developing angle closure glaucoma (*Wang et al., 2019*; *Crahay et al., 2021*). Likewise, keratoconus is associated with a larger corneal diameter and changes in anterior chamber parameters (*Crahay et al., 2021*; *Riva et al., 2020*). Without comparative data from these patient groups, we cannot determine whether the relationships we observed between ocular biometry and demographic factors remain consistent in those with eye diseases. Future studies should include individuals with common anterior segment disorders to better understand the clinical value of biometric measurements in disease detection and management.

## CONCLUSIONS

The WTW corneal diameter and AC parameters provided in the current study would aid as a reference for ophthalmologists and other eye care professionals to precisely diagnose ocular diseases such as angle closure glaucoma and corneal disorders. These values also aid surgeons in calculating IOL and ICL size for cataract and refractive surgeries. Large sample Future studies with diverse population groups and different ocular diseases are required to validate our initial findings.

### Funding
This work was supported by Deanship of Scientific Research and the College of Applied Medical Sciences Research Center at King Saud University. The funders had no role in study design, data collection and analysis, decision to publish, or preparation of the manuscript.

### Grant Disclosures
The following grant information was disclosed by the authors:
Deanship of Scientific Research and the College of Applied Medical Sciences Research Center at King Saud University.

### Competing Interests
The authors declare that they have no competing interests.

### Author Contributions
- Wafa M. Alotaibi conceived and designed the experiments, performed the experiments, analyzed the data, authored or reviewed drafts of the article, and approved the final draft.
- Naveen Challa analyzed the data, prepared figures and/or tables, authored or reviewed drafts of the article, and approved the final draft.
- Saif Hassan Alrasheed analyzed the data, prepared figures and/or tables, authored or reviewed drafts of the article, and approved the final draft.
- Rahaf Nasser Abanmi performed the experiments, analyzed the data, authored or reviewed drafts of the article, and approved the final draft.

## Human Ethics

The following information was supplied relating to ethical approvals (*i.e.*, approving body and any reference numbers):

King Saud University Medical City's ethical committee, E-21-6448.

## Data Availability

The raw data are available in the Supplemental File.

## Supplemental Information

Supplemental information for this article can be found online at http://dx.doi.org/10.7717/peerj.19227#supplemental-information.

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
