# Peer review of "Measurements of white-to-white corneal diameter and anterior chamber parameters using the Pentacam AXL wave and their correlations in the adult Saudi population"

_PeerJ, doi:10.7717/peerj.19227_

## Round 0.1 · original submission · Minor Revisions

Both reviewers have a number of concerns, however they should all be possible to address

·

Basic reporting

Generally good, some sentences need expanding for clarity as detailed in the appropriate section below.

Experimental design

The study seems confused as to the aspect of age in the experimental design. In the abstract the state that the study examines “the Saudi adult population”, however in the introduction the authors state that the study will examine the “young Saudi population” (lines 63-66).
The paper actually examines an age range from 18-60 years, however the age range is skewed substantially towards younger ages (>half of the subjects are in the 18-25 years range) which will affect any correlation data where age is used as one of the variables.
As the authors point out themselves, while the data can be considered reliable, with the age range and number of subjects, caution should be exercised with extending the results to the whole population, particularly at the older age ranges, as there is insufficient numbers pf subjects.
The statement “The Pentacam AXL Wave is the gold standard……the eye” is not justified scientifically, and actually sounds like it has been pulled directly from the manufacturer’s blurb. I am sure the instrument has been evaluated and compared against a range of other instruments which carry out the same measurements, but I can see no reason to ascribe the term “gold standard” to this or any other anterior segment imaging system, unless there are studies which show that it gives the closest values to the true measurement of these parameters, which is unlikely.
The text states that the measurements were taken on the right eye twice (lines 106-109).
The authors should say which parameters they think are likely to be affected by diurnal variation, as I doubt WTW is one of them (lines 106-107).

Validity of the findings

The findings are valid, subject to the following caveats:
a. The MSE has not been calculated correctly. The MSE is the arithmetic sum of sphere + ½ cyl, but the authors seem not to have done this consistently, e.g. in line 2 of the raw data, the Rx is -1.75/-0.75 which should result in a MSE of 2.125, but the authors seem to have rounded this to 2.25. This could affect the value of any correlations and should be corrected.
b. The authors report the distributions of certain parameters but not AL, Rx and age, yet they use these parameters in the data analysis. They should report the distributions of these parameters also and the normality of these parameters. The age distribution is almost certainly not normal, as it is skewed towards younger ages.
c. The skewed distribution of age towards younger ages will almost certainly affect the correlations of parameters with age, and makes the correlations less reliable especially when considering older age groups which are less prevalent in the data.
d. The authors should not present non-significant correlations without saying that they are not significant.
e. The authors describe correlations as high, moderate or poor, but there is no guidance as to the choice of such terms, nor what that actually means in the context of the data.
f. Using Rx as a measure in some of the correlations should be avoided, as Rx is basically a corollary of AL which is the primary determinant of refractive error. I would tend to use AL rather than Rx and would probably not use both.

Additional comments

a. This is a useful paper which presents data on anterior eye parameters in a specific racial group which is under-represented in the literature.
The authors have correctly identified the subject numbers as a limitation of the data.
b. Discussion
i. Lines 153-154: what does “with varying degrees of success” mean? I assume these published papers all measured WTW dimension accurately.
ii. Line 155: Change “higher than that of” to “higher than that reported in”
iii. Line 156: change “individuals” to groups
iv. Line 158: what does “utilizing a distinct set of measurements” mean?
v. Line 160-161”Pentacam seems to” this is very unclear. Was the difference significant and if so, by how much?
vi. Line 163-164: “corneal diameter in our study are” change to “corneal diameter in our study may be”
vii. Line 163-164: Are you saying that the use of the Pentacam explains the fact that WTW is larger than in other studies or not? Can you say what the data looks like if corrected for use of Pentacam?
viii. Lines 181-184: Is the difference in ACD related in any way to differences in age distributions between studies? If the differences could be due to using different instruments, you need to say what the instruments were and why there might be a difference.
ix. Line 187-188: “This variance may be due to the use of different instruments and racial disparities.” This needs to be elucidated, can you say how much each factor contributes to the difference.

Reviewer 2 ·

Basic reporting

1) While the literature cited is robust, the manuscript could benefit from additional background on population-specific variations in these parameters. A more detailed discussion of why these measurements are particularly relevant to the Saudi population would enhance the study's context.

2) Consider elaborating on how these results compare directly to other population studies and their implications on clinical practices.

3) Emphasize any limitations or suggest follow-up research that could extend the findings without fragmenting the publication into smaller, related studies.

Experimental design

1) Reinforce the unique contribution of the study to the broader field by explicitly comparing these findings to those in populations where such data is more readily available, such as Western or Asian cohorts.

2) While the use of Pentacam is highlighted, briefly explaining why this tool is the most appropriate choice for the measurements would strengthen the rationale for its use.

Validity of the findings

1) While the statistical tests are sound, consider including a brief statement on potential sources of bias or variance, such as environmental factors or inter-operator variability, to strengthen the robustness of findings.

2) Include a more explicit statement on the benefit of replicating these findings in diverse populations or using alternate biometric tools to validate and compare results across studies. This would underline the broader relevance and encourage similar studies to build a comprehensive database of ocular biometry across different ethnicities.

3) To reinforce the validity of the conclusions, consider adding a statement on the potential clinical impact of implementing these reference values in daily practice or as a baseline for future studies.

---

## Round 0.2 · Minor Revisions

Dear authors,

Thank you for your submission. The peer review process has provided valuable feedback, and we appreciate the work you’ve put into the manuscript. However, before we can proceed to production, I request that you address several minor areas to ensure clarity and accuracy in your work.

Your research employs a prospective cross-sectional design, however to strengthen your conclusions consider to elaborate on how this design supports your study objectives and complete any limitations inherent that may affect the interpretation of your findings. Integrating longitudinal data could provide insights into causal relationships and temporal changes. If not included, please justify this decision and discuss its implications.
The inclusion of 82 participants may be limited, depending on your statistical analyses. Conducting a power analysis can help determine if this sample size is sufficient to detect significant differences, especially when stratifying by variables such as sex or age.
The study focuses on healthy individuals but lacks a control group with specific ocular conditions. Discuss its impact on the generalization of your results.
Excluding participants with conditions like diabetic retinopathy, corneal opacity, and prior ocular surgeries may limit the applicability of your findings. Please provide a rationale for these exclusions and discuss how they might influence the study's outcomes. (basically, thus far clearly defining your hypothesis, objectives and aims) . While the use of tools like the Pentacam AXL Wave is appropriate, please provide comprehensive details on the equipment calibration and the training received by personnel conducting the examinations. This information is crucial for ensuring reproducibility.
A more detailed examination of how demographic variables (e.g., age, sex, refractive errors) relate to the measured outcomes would be beneficial. Please specify the statistical methods used to control for potential confounding factors.
I think it is possible and beneficial for impact and significance to report a more thorough analysis and discussion of the results, including statistical significance and the implications of your findings. Linking your results to existing literature and suggesting future directions. (also, make sure that all acronyms have been defined; and you may need to consider increase font size of the graphs axis for readability. In the figures, also consider to mention the key "takehome" message of the data presented in the legend).

After addressing these points, i believe your manuscript will make a more substantial contribution to the understanding of ocular measurements in the Saudi population and their clinical relevance. Look forward to receiving the revised manuscript.

Reviewer 2 ·

Basic reporting

The authors have thoroughly addressed all of my comments.

Experimental design

The authors have thoroughly addressed all of my comments.

Validity of the findings

The authors have thoroughly addressed all of my comments.

Additional comments

The authors have thoroughly addressed all of my comments.

---

## Round 0.3 · Minor Revisions

Dear authors, thank you for your resubmission. I have reviewed your manuscript and the accompanying rebuttal. While the rebuttal has clarified several aspects, there are some minor issues that require your revision before acceptance and production. These revisions aim to further clarify certain points, prospect potential questions from readers, and improve the overall presentation of your work.
Include in the manuscript the discussion to address the potential impact of lacking a control group with specific ocular conditions (ie, as asked before, consider how the absence of a control group (with conditions like glaucoma or keratoconus) might affect the clinical applicability of your results, in the manuscript (this has not have been found in the tracked changes document) .

The response to the suggestion for a more thorough analysis of the results was noted. I would like if possible that you more explicitly link your results to existing literature, and highlight the the implications of your findings.
In the tables and figures' legends, please also ensure acronyms are defined;

Table 1: The legend currently repeats the description unnecessarily. Please revise to avoid repetition and ensure that the most significant results are highlighted in the legend.

Table 2: The legend includes redundant phrasing. Please eliminate the repetition and clearly define all acronyms. The key message of the table should be stated clearly in the legend.

Table 3: The current legend is quite detailed, which is good, but it could be improved by emphasizing the most important findings. Highlight key results in the legend.

Figure 1: Ensure all acronyms are defined in the figure and improve the legend with more detailed information about each sub-figure (a, b, c). highlight the key takeaways from the figure.

Figures 2 and 3: The "Figure 2" and "Figure 3" labels are appropriate, but the legends for these figures still need improvement. Follow the guidance provided for the other figures, emphasizing key information that is essential for understanding the data.

Once these adjustments are made, I believe your paper will be ready for acceptance.

Thank you for your hard work and attention to these details.

---

## Round 0.4 · accepted · Accept

Dear authors,
Congratulations! I am now accepting your manuscript for publication, as all issues have been resolved. Thank you for your contribution!